# Does pre-emptive dexamethasone provide prophylaxis against sugammadex-induced bradycardia? A retrospective study

Jonathan S. Jahr [1]☯*, Pamela A. Chia[1]☯, Tristan Grogan[2], Phiona Nansubuga[3], Julia Vogt[4], Victoria Klinewski[4], Thomas J. Ebert[4]

**1** Department of Anesthesiology and Perioperative Medicine, David Geffen School of Medicine at UCLA, Los Angeles, California, United States of America, **2** Department of Medicine Statistics Core, David Geffen School of Medicine at UCLA, Los Angeles, California, United States of America, **3** Department of Anesthesiology, Uganda Cancer Institute, Kampala, Uganda, **4** Department of Anesthesiology, Zablocki VA Medical Center, Medical College of Wisconsin, Milwaukee Wisconsin, United States of America

☯ These authors contributed equally to this work.
* jsjahr@ucla.edu

## Abstract

Sugammadex is a cyclodextrin used to reverse neuromuscular block with amino-steroid nondepolarizing muscle relaxants, rocuronium and vecuronium. Sugammadex-induced bradycardia was recently demonstrated in a single-blind, placebo-controlled study in patients receiving rocuronium for neuromuscular block. It has also been hypothesized that the bradycardia and rare instances of cardiac arrest occurring after the use of sugammadex may be due to a transient decrease in circulating corticosteroids, causing a temporary 'mini Addisonian crisis.' It was proposed that the administration of corticosteroids such as dexamethasone for post-operative nausea and vomiting (PONV) management might offer prophylaxis against these adverse occurrences. The study database was queried from a prospective study on sugammadex-related bradycardia, which was approved by the Human Studies Review Board and exempt from patient consent requirements. Patients were grouped into those that had or had not received dexamethasone as prophylaxis for PONV prior to the administration of sugammadex, and heart rate changes were evaluated 5 minutes after sugammadex administration. A total of 103 subjects were evaluated, of whom 38 received intravenous dexamethasone (either 4 mg, 8 mg, or 10 mg) during their anesthetic course and 65 patients had not received dexamethasone. The average heart rate (HR) slowing (3.2 bpm ± 3.9 in the control group, 3.7 bpm ± 3.8 in the dexamethasone group), and maximal HR slowing (5.0 bpm ± 3.9 in the control group, 5.0 bpm ± 3.8 in the dexamethasone group) over the five minutes following sugammadex administration were not significant between groups (average HR slowing p = 0.553, maximal HR slowing p = 0.988). These results potentially negate the proposed theory, or it may be that corticosteroids with more mineralocorticoid activity

**Data availability statement:** All relevant data are within the paper and its Supporting Information files.

**Funding:** The author(s) received no specific funding for this work.

**Competing interests:** The authors have declared that no competing interests exist.

such as fludrocortisone or hydrocortisone are required to prevent this effect. Larger studies or prospective trials evaluating this effect with cortisol concentration measurement are needed to further evaluate the hypothesis.

## Introduction

Sugammadex is a selective relaxant binding agent that antagonizes neuromuscular block with amino-steroid non depolarizing muscle relaxants rocuronium and vecuronium [1]. Adverse effects of sugammadex have been well documented, including bradycardia, defined as a heart rate (HR) less than 60 bpm, and asystole [2]. Recently, a blinded study reported a consistent HR slowing effect from sugammadex, on occasion that was clinically significant to require rescue therapy [3]. The relationship between perioperative beta-blocker use and sugammadex-related bradycardia has yet to be investigated.

Jahr and colleagues hypothesized that sugammadex-induced bradycardia may be due to a transient decrease in circulating corticosteroids, causing a temporary 'mini Addisonian crisis.' This is thought to resolve when sugammadex preferentially binds to neuromuscular blockers, thus reducing corticosteroid binding in the bloodstream [4,5]. This theory is based on the binding affinity of sugammadex. Sugammadex has a high binding affinity for steroidal neuromuscular blocking agents, such as rocuronium (Ka of $1.79 \times 10^7$ mol/L) and vecuronium (Ka of $5.72 \times 10^6$ mol/L) [6,7]. Comparatively, the binding affinity to corticosteroids is relatively weak, as indicated by the Ka values: dexamethasone ($<1.00 \times 10^3$ mol/L), hydrocortisone ($5.48 \times 10^4$ mol/L), and fludrocortisone ($1.00 \times 10^5$ mol/L) [6,7]. When administered, sugammadex may initially bind to steroids in the central circulation. In patients predisposed to bradycardia due to reduced corticosteroid concentrations, this initial binding could theoretically contribute to the effect, and may explain reports of cardiac arrest refractory to catecholamines and anticholinergics. However, as sugammadex circulates to peripheral sites, including muscle vascular beds, it would preferentially bind to rocuronium or vecuronium. This binding would subsequently release corticosteroids back into the circulation, potentially reducing the risk of more severe bradycardia.

Dexamethasone in doses of 4–10 mg in adult patients is commonly used for prophylaxis of post-operative nausea and vomiting (PONV) in patients undergoing general anesthesia [8]. Theoretically, if dexamethasone is administered before sugammadex, and if sugammadex-induced bradycardia is due to a decrease in circulating corticosteroids transiently, then dexamethasone may offer prophylaxis against the bradycardia or possibly make it easier to treat [4,5].

## Methods

This study retrospectively evaluated patients to determine if dexamethasone administered prior to sugammadex for routine PONV prophylaxis ameliorated sugammadex-induced bradycardia or made it potentially less refractory to standard therapy. The study database was queried from a prospective study on sugammadex-related

bradycardia on November 29, 2022, and all data was de-identified [3]. The prospective study investigated HR responses to sugammadex antagonism. Importantly, the study demonstrated that sugammadex, when used for muscle relaxant reversal, may rapidly decrease HR, although the degree of HR reduction varied. The same published data set was used for the retrospective analysis. The evaluation was approved by the Human Studies Review Board and exempt from patient consent requirements. Patients were grouped into those that had or had not received dexamethasone as prophylaxis for PONV prior to the administration of sugammadex, and heart rate changes were evaluated 5 minutes after sugammadex administration. The primary aim of this study was to determine whether dexamethasone attenuated HR slowing. Second, this study retrospectively analyzed the same cohort to determine if perioperative beta-blocker use effected sugammadex-related bradycardia, evaluating the relationship between beta-blocker use and the percentage of bradycardic episodes.

## Statistical methods

Patient characteristics and study variables were compared between groups using means (standard deviations) or frequency (percentage), unless otherwise noted. Differences between groups were formally assessed using the t-test or chi-square test. HR measures/changes between groups were compared using linear regression models with terms for group and beta blocker usage due to the potential confounding issue of beta blockers being used less frequency in the dexamethasone group. Statistical analyses were run using R V4.1.0 (Vienna, AU, www.r-project.org) and p-values less than 0.05 were considered statistically significant.

## Power and sample size calculation

An a priori sample size calculation was not computed for this study. The sample size was determined based on a 3 month period (May to September 2021) when two medical students (VBK and MTA) were available to assist with data collection. After determining the available sample size, a retrospective analysis computed the effect sizes that could be detected with adequate power (>80%). It was determined that a sample size of 103 provided adequate power (>80%) to detect effect sizes as small as 30% for binary outcomes (e.g., p1 = 35%, p2 = 65%, two-tailed alpha = 0.05, group weights 2:1 using the chi-square test). For continuous outcomes, this sample size provided adequate power (>80%) to detect effect sizes (Cohen's d) as small as 0.6 between groups (two-sample t-test, two-tailed alpha = 0.05, group weights 2:1).

## Results

The study database consisted of 103 subjects, of whom 38 received intravenous dexamethasone (either 4 mg, 8 mg, or 10 mg) during their anesthetic course prior to sugammadex and 65 patients were identified as not receiving dexamethasone. The two groups were similar in terms of demographics and co-morbidities (Table 1). The average heart slowing (3.2 bpm ± 3.9 in the control group, 3.7 bpm ± 3.8 in the dexamethasone group), and maximal HR slowing (5.0 bpm ± 3.9 in the control group, 5.0 ± 3.8 in the dexamethasone group) over the five minutes following sugammadex administration was determined in each group (Table 2). Although the dexamethasone group showed a slightly greater average HR slowing (3.7 bpm ± 3.8 vs 3.2 ± 3.9 in the control group), this difference was not statistically significant (p = 0.553, Fig 1). The maximal HR slowing was also not significant between groups (p = 0.988, Fig 1). After adjusting for baseline patient characteristics, the adjusted p-values were 0.570 and 0.907, respectively.

As an exploratory analysis, we evaluated the relationship between dexamethasone dose and our outcomes using Spearman correlations. Among the treatment group, no significant dose-response relationships were observed (dose vs. average HR change (ρ=0.06, p = 0.741), dose vs. maximal HR slowing (ρ=0.03, p = 0.847). To further explore this, we assigned a dose of 0 mg to the control group and reran the analysis. The results similarly showed no significant associations (dose including controls vs. average HR change: ρ=-0.06, p = 0.551, dose including controls vs. maximal HR slowing: ρ=0.01, p = 0.950).

**Table 1. Patient characteristics.**

|  | No Dexamethasone | Dexamethasone | Overall | *p* value |
|---|---|---|---|---|
|  | (N = 65) | (N = 38) | (N = 103) |  |
| **Age** | 67.5 (10.6) | 64.9 (10.1) | 66.5 (10.5) | 0.238 |
| **BMI** (kg/m²) | 30.1 (7.3) | 28.9 (5.0) | 29.6 (6.6) | 0.398 |
| **Race** |  |  |  | 0.101 |
| White | 48 (76.2%) | 31 (91.2%) | 79 (81.4%) |  |
| African American | 15 (23.8%) | 3 (8.8%) | 18 (18.6%) |  |
| Other | 0 | 0 | 0 |  |
| **ASA Score** |  |  |  | 0.299 |
| 1 | 1 (1.6%) | 2 (5.3%) | 3 (2.9%) |  |
| 2 | 18 (28.1%) | 15 (39.5%) | 33 (32.4%) |  |
| 3 | 44 (68.8%) | 21 (55.3%) | 65 (63.7%) |  |
| 4 | 1 (1.6%) | 0 | 1 (1.0%) |  |
| **Comorbidities** |  |  |  |  |
| HTN | 42 (67.7%) | 20 (52.6%) | 62 (62.0%) | 0.131 |
| DM | 20 (31.7%) | 9 (23.7%) | 29 (28.7%) | 0.386 |
| CAD | 9 (14.5%) | 2 (5.3%) | 11 (11.0%) | 0.198 |
| Malignancy | 19 (30.6%) | 9 (23.7%) | 28 (28.0%) | 0.452 |
| **Outpatient Medications** |  |  |  |  |
| Beta-blockers | 18 (27.7%) | 4 (10.5%) | 22 (21.4%) | 0.04 |
| Insulin | 8 (12.3%) | 4 (10.5%) | 12 (11.7%) | 1 |
| ACEI/ARBs | 21 (32.3%) | 13 (34.2%) | 34 (33.0%) | 0.843 |
| **Dexamethasone Dose** |  |  |  | -- |
| 4mg | -- | 9 (8.7%) | -- |  |
| 8mg | -- | 17 (16.5%) | -- |  |
| 10mg | -- | 12 (11.7%) | -- |  |

ᵃValues are reported as mean (SD) for age and BMI, frequency (%) for race, ASA score comorbidities, and outpatient medications. ASA, American Society of Anesthesiologists; BMI, body mass index; HTN, hypertension; DM, diabetes mellitus; CAD, coronary artery disease;, ACEI, angiotensin-converting enzyme inhibitor; ARB, Angiotensin II receptor blocker.

**Table 2. Study outcomes.**

|  | Overall | No Dexamethasone | Dexamethasone | *p* value |
|---|---|---|---|---|
|  | (N = 103) | (N = 65) | (N = 38) |  |
| **Baseline average** | 65.9 (10.2) | 67.1 (10.4) | 63.9 (9.7) | 0.131 |
| **Average HR change** | -3.4 (3.9) | -3.2 (3.9) | -3.7 (3.8) | 0.553 |
| **Maximal HR slowing** | -5.0 (3.9) | -5.0 (3.9) | -5.0 (3.8) | 0.988 |

ᵃValues reported as mean (SD). HR, heart rate.

The dexamethasone group had significantly fewer patients on beta blockers as an outpatient medication (10.5% vs 27.7%, p = 0.04) and was adjusted for in our secondary analysis, where the effect of perioperative beta-blocker use was included in the models. We also evaluated the incidence of bradycardia (defined as HR less than 60 bpm). In the overall cohort, 38.3% (31/81) of subjects not on beta-blockers experienced bradycardia compared to 59.1% (13/22) of those on beta-blockers. This difference was not statistically significant (p = 0.080). Among patients on beta-blockers, the subgroup

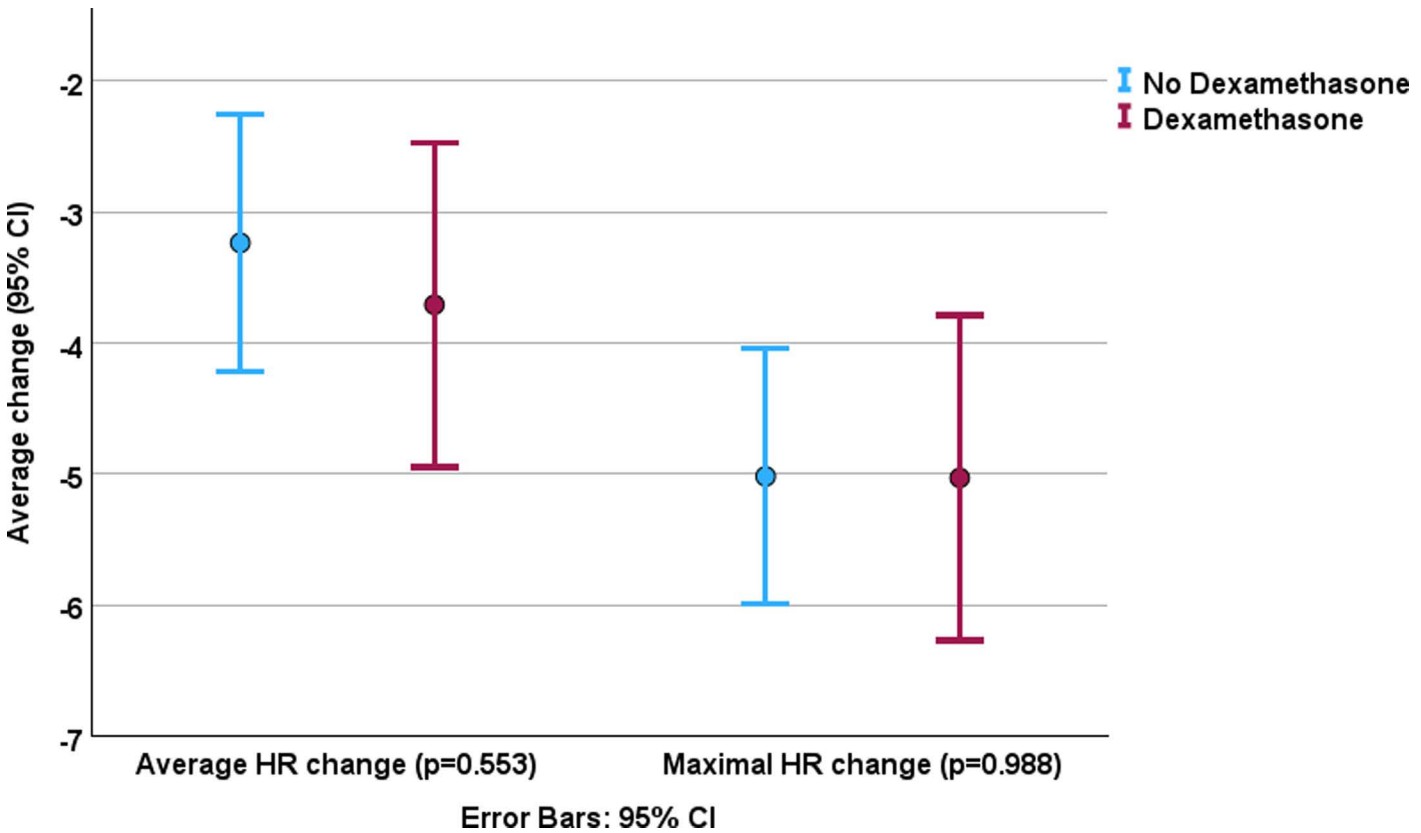

**Fig 1.  The average and maximal heart rate slowing over five minutes following sugammadex administration in the control and dexamethasone groups.**

analysis revealed that bradycardia occurred in 29.8% (14/47) of those who did not receive dexamethasone and 50.0% (17/34) of those who did, with this difference approaching statistical significance (p = 0.065). Adjusted analyses accounting for beta-blocker use yielded p-values that were consistent with the unadjusted analyses. Adjusted analyses confirmed that beta-blocker usage did not confound the primary results related to heart rate changes, as the adjusted p-values were consistent with the unadjusted analyses.

## Discussion

Dexamethasone is a long-acting synthetic corticosteroid that is commonly administered intravenously in the perioperative setting. Dexamethasone is used to reduce PONV as well as decrease post-operative pain, potentially from the anti-inflammatory properties of glucocorticoids [9]. No significant differences in heart rate changes between the patients that received dexamethasone as prophylaxis for PONV were noted compared to those that did not. As an exploratory analysis, we evaluated the relationship between dexamethasone dose and our outcomes using Spearman correlations, and our findings suggest that differences in dexamethasone dose did not significantly influence the observed outcomes. The lack of effect observed with dexamethasone may either negate the theory proposed by Jahr and colleagues [4,5], or it may also be that other corticosteroids, with more mineralocorticoid activity, such as hydrocortisone or fludrocortisone, are required to prevent this effect [10]. Dexamethasone, notably only has glucocorticoid activity and negligible mineralocorticoid activity. In comparison, fludrocortisone has potent mineralocorticoid activity, and hydrocortisone, equal amounts

of mineralocorticoid and glucocorticoid activity [10]. Prospective trials evaluating this effect with cortisol concentration measurement are needed to prove or disprove this hypothesis.

In addition, despite bradycardia reported as an adverse effect of this muscle relaxant in several reports [3], recommendations for patients on perioperative beta-blockers has yet to be elucidated. It was notable that the rate of bradycardia was 59.1% in patients on beta blockers, and 38.3% in patients not on beta blockers (p = 0.08). While the clinical significance of the bradycardic episodes was not investigated, as sugammadex use grows exponentially, the concerns about the episodes may grow. This has clinical implications as bradycardia can lead to hypotension, particularly in the context of relative hypovolemia. It may also cause ventricular stretching and increase oxygen consumption due to the work required for contraction. Thus the authors encourage a prospective evaluation with a larger sample size to gain insights in the potential interactions between perioperative beta blocker use and sugammadex. Furthermore, the small sample size was a limitation of the study. The retrospective calculation suggested that this study was not adequately powered to detect the smaller observed effect sizes (0.12 for average HR change and 0.003 for max HR slowing). As such, the ability to detect significant differences for these specific outcomes was limited by the small observed effect sizes in our dataset.

## Conclusion

In conclusion, the incidence of bradycardia refractory to standard treatment is rare though this may increase as the utilization of sugammadex increase [4]. It is important to raise awareness among clinicians about the potential for a cross-reaction in patients receiving beta blockers and sugammadex. Vigilance during and after administration of sugammadex is always recommended [11]. Once the etiology of sugammadex-induced bradycardic episodes is elucidated, hopefully a specific therapy will be recommended to treat to symptomatic patients.

## Supporting information

**S1 File. Study Data.**
(XLSX)

## Author contributions

**Conceptualization:** Jonathan S. Jahr, Pamela A. Chia, Tristan Grogan, Phiona Nansubuga, Thomas J. Ebert.

**Data curation:** Pamela A. Chia, Tristan Grogan, Phiona Nansubuga, Julia Vogt, Victoria Klinewski, Thomas J. Ebert.

**Formal analysis:** Jonathan S. Jahr, Pamela A. Chia, Tristan Grogan, Julia Vogt, Victoria Klinewski, Thomas J. Ebert.

**Investigation:** Jonathan S. Jahr, Phiona Nansubuga, Thomas J. Ebert.

**Methodology:** Jonathan S. Jahr, Tristan Grogan, Thomas J. Ebert.

**Project administration:** Jonathan S. Jahr.

**Supervision:** Jonathan S. Jahr, Thomas J. Ebert.

**Validation:** Thomas J. Ebert.

**Writing – original draft:** Jonathan S. Jahr, Pamela A. Chia, Tristan Grogan, Phiona Nansubuga, Thomas J. Ebert.

**Writing – review & editing:** Jonathan S. Jahr, Pamela A. Chia, Tristan Grogan, Phiona Nansubuga, Julia Vogt, Victoria Klinewski, Thomas J. Ebert.

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
