## [Decision Letter · Decision Letter 0]

11 Dec 2024

PONE-D-24-42641Does Pre-emptive Dexamethasone Provide Prophylaxis Against Sugammadex-Induced Bradycardia? A Retrospective StudyPLOS ONE

Dear Dr. Jahr,

Thank you for submitting your manuscript to PLOS ONE. After careful consideration, we feel that it has merit but does not fully meet PLOS ONE’s publication criteria as it currently stands. Therefore, we invite you to submit a revised version of the manuscript that addresses the points raised during the review process.

We look forward to receiving your revised manuscript.

Kind regards,

Mohammed Misbah Ul Haq, Pharm-D

Academic Editor

PLOS ONE

Reviewers' comments:

Reviewer's Responses to Questions

**Comments to the Author**

1. Is the manuscript technically sound, and do the data support the conclusions?

Reviewer #1: Partly

Reviewer #2: No

2. Has the statistical analysis been performed appropriately and rigorously? 

Reviewer #1: Yes

Reviewer #2: I Don't Know

3. Have the authors made all data underlying the findings in their manuscript fully available?

Reviewer #1: Yes

Reviewer #2: Yes

4. Is the manuscript presented in an intelligible fashion and written in standard English?

Reviewer #1: Yes

Reviewer #2: Yes

5. Review Comments to the Author

Reviewer #1: Please find attached the review comments to the author

Overall, the manuscript presents an interesting and relevant study. The topic of sugammadex-induced bradycardia and its interaction with dexamethasone and beta-blockers is important, but the findings in this study suggest that the hypotheses might need further refinement. The suggestions provided above will help improve clarity, precision, and overall readability. Good work, and I encourage further exploration of this topic!

Reviewer #2: 1. It is unclear how the sample size was calculated. It seems like it just reflects the number of patients that were found on the database rather than the number of participants required to detect a difference in the results. I would recommend a statistician review the sample size calculation and the statistical analyses. Please confirm the primary outcome and whether the sample size was powered on this.

2. There is no mention of the dose of dexamethasone that these participants received, or when or how it was administeredt (beyond the mention of 4-8mg in the Background section and that they had received this prior to sugammedex). Please include further details on this.

3. Please confirm the p value for the change in maximal HR (in the table it is listed as p0.988 but in the text it is 0.23)

4. I am unsure the sample size in this study is large enough to support or negate the original hypothesis, as the authors state in the Discussion.

5. Please include the results of the secondary analyses (i.e. effect of peri-operative beta-blockers on bradycardia) in the Results section. How many participants experienced bradycardia after sugammedex? How was bradycardia defined?

6. PLOS authors have the option to publish the peer review history of their article (what does this mean? ). If published, this will include your full peer review and any attached files.

**Do you want your identity to be public for this peer review?** For information about this choice, including consent withdrawal, please see our Privacy Policy .

Reviewer #1: No

Reviewer #2: No

---

## [Author Response · Author response to Decision Letter 1]

28 Jan 2025

Response to Reviewers

Manuscript: PONE-D-24-42641

Does Pre-emptive Dexamethasone Provide Prophylaxis Against Sugammadex-Induced Bradycardia? A Retrospective Study

The authors appreciate the Academic Editor’s and Reviewers’ helpful comments, which we have addressed below. Academic Editor and Reviewer comments are provided below in italics, with our subsequent response in blue.

Academic Editor:

Response: We thank you for this guidance and have updated the manuscript to meet PLOS ONE’s style requirements, including those for file naming.

Response: We thank you for this recommendation and will make our data freely accessible if accepted for publication. The authors agree to place the data under Supplemental Information as “S1 File”.

Response: We thank the editor for bringing this to our attention. We have amended the abstract on the online submission form to match the abstract in the manuscript.

Reviewers:

1. Is the manuscript technically sound, and do the data support the conclusions?

Reviewer #1: Partly

Response: We thank the reviewer for this comment. We sincerely believe that the data presented in this manuscript is derived from rigorously conducted experiments. The study's dataset originates from a prospective investigation into sugammadex-related bradycardia (PMID: 35759402, DOI: 10.1213/ANE.0000000000006131), which was published in Anesthesia & Analgesia in 2022.

Reviewer #2: No

Response: Thank you for this comment. This retrospective study was conducted to explore a published hypothesis regarding a potential mechanism for sugammadex-related bradycardia (PMID: 36608072, DOI: 10.1097/MJT.0000000000001590). Although the results did not support the hypothesis, we believe this study is important to share because sugammadex is widely administered during anesthesia, and the cause of sugammadex-induced bradycardia remains unknown, with possibly clinically significant implications.

2. Has the statistical analysis been performed appropriately and rigorously?

Reviewer #1: Yes

Response: We thank the reviewer for this positive comment.

Reviewer #2: I Don't Know

Response: We thank the reviewer for this comment. We appreciate your inquiry regarding the appropriateness and rigor of the statistical analysis. The statistical methods used in this study were carefully selected and conducted by a biostatistician with over 10 years of experience (Master’s-level training). The analyses were performed in accordance with standard statistical practices and guidelines, ensuring robust and reliable results.

3. Have the authors made all data underlying the findings in their manuscript fully available?

Reviewer #1: Yes

Response: We thank the reviewer for this positive comment.

Reviewer #2: Yes

Response: We thank the reviewer for this positive comment.

4. Is the manuscript presented in an intelligible fashion and written in standard English?

Reviewer #1: Yes

Response: We thank the reviewer for this positive comment.

Reviewer #2: Yes

Response: We thank the reviewer for this positive comment.

5. Review Comments to the Author

Reviewer #1: Please find attached the review comments to the author

Overall, the manuscript presents an interesting and relevant study. The topic of sugammadex-induced bradycardia and its interaction with dexamethasone and beta-blockers is important, but the findings in this study suggest that the hypotheses might need further refinement. The suggestions provided above will help improve clarity, precision, and overall readability. Good work, and I encourage further exploration of this topic!

Response: We thank the reviewer for the positive feedback. We believe this study is important to share because sugammadex is widely administered during anesthesia, yet the cause of sugammadex-induced bradycardia remains unknown. Our aim is to address this gap by testing our hypothesis through this retrospective analysis. Publishing this report will help disseminate current knowledge, encourage other groups to explore the question prospectively, and investigate potential relationships.

Reviewer #2: 1. It is unclear how the sample size was calculated. It seems like it just reflects the number of patients that were found on the database rather than the number of participants required to detect a difference in the results. I would recommend a statistician review the sample size calculation and the statistical analyses. Please confirm the primary outcome and whether the sample size was powered on this.

Response: The sample size for this study was determined based on logistical constraints, specifically the time period during which two medical students (VBK and MTA) were available to assist with data collection (May to September 2021). While the sample size was not chosen prospectively based on a formal power calculation for a specific primary outcome, we retrospectively computed the effect sizes that could be detected with adequate power (>80%) for the available sample size.

For binary outcomes, this sample size provides adequate power to detect differences as small as 30% (e.g., p1=35%, p2=65%; two-tailed alpha=0.05, group weights 2:1) using a chi-square test. For continuous outcomes, we computed that the sample size could detect effect sizes (Cohen’s d) as small as 0.6 with adequate power (>80%) using a two-sample t-test with the same alpha and group weights.

While the primary outcomes of interest were average HR change and max HR slowing, the retrospective calculation suggests that the study was not adequately powered to detect the smaller observed effect sizes (0.12 for HR change and 0.003 for max HR slowing). As such, the ability to detect significant differences for these specific outcomes was limited by the small observed effect sizes in our dataset. Additional detail about the sample size calculation were added under “Methods” (paragraph 3) and the limitations of our sample size were acknowledged under “Discussion” (paragraph 2).

Methods: Power and sample size calculation

“An a priori sample size calculation was not computed for this study. The sample size was determined based on a 3 month period (May to September 2021) when two medical students (VBK and MTA) were available to assist with data collection. After determining the available sample size, a retrospective analysis computed the effect sizes that could be detected with adequate power (>80%). It was determined that a sample size of 103 provided adequate power (>80%) to detect effect sizes as small as 30% for binary outcomes (e.g. p1=35%, p2=65%, two-tailed alpha=0.05, group weights 2:1 using the chi-square test). For continuous outcomes, this sample size provided adequate power (>80%) to detect effect sizes (Cohen’s d) as small as 0.6 between groups (two-sample t-test, two-tailed alpha=0.05, group weights 2:1).”

Discussion:

“Furthermore, the small sample size was a limitation of the study. The retrospective calculation suggested that this study was not adequately powered to detect the smaller observed effect sizes (0.12 for average HR change and 0.003 for max HR slowing). As such, the ability to detect significant differences for these specific outcomes was limited by the small observed effect sizes in our dataset.”

2. There is no mention of the dose of dexamethasone that these participants received, or when or how it was administered (beyond the mention of 4-8mg in the Background section and that they had received this prior to sugammedex). Please include further details on this.

Response: We thank the reviewer for the suggestion to expand on the specifics of the dosing and administration of dexamethasone. We reviewed our records and determined the dexamethasone doses administered to the 38 patients in the treatment group. The dose of dexamethasone administered intravenously was either 4 mg, 8 mg, or 10 mg. This information was added under “Results” (paragraph 1). This breakdown of the administered doses was also added to Table 1 and was as follows:

• 4 mg: 9 patients (8.7%)

• 8 mg: 17 patients (16.5%)

• 10 mg: 12 patients (11.7%)

Results:

The study database consisted of 103 subjects, of whom 38 received intravenous dexamethasone (either 4 mg, 8 mg, or 10 mg) during their anesthetic course prior to sugammadex and 65 patients were identified as not receiving dexamethasone.

Table 1:

3. Please confirm the p value for the change in maximal HR (in the table it is listed as p0.988 but in the text it is 0.23)

Response: We thank the reviewer for the recommended edit for clarity. The p-value for the change in maximal heart rate is 0.998 (Fig 1, Table 2) and 0.907 in the adjusted model. The adjusted model accounts for all variables in Table 1, including race, age, BMI, ASA classification, hypertension, diabetes, coronary artery disease, malignancy, outpatient beta-blocker medications, insulin use, and ACEI/ARBs. The text has been updated to reflect this under Results (paragraph 1).

“The average heart slowing (3.2 bpm ± 3.9 in the control group, 3.7 bpm ± 3.8 in the dexamethasone group), and maximal HR slowing (5.0 bpm ± 3.9 in the control group, 5.0 ± 3.8 in the dexamethasone group) over the five minutes following sugammadex administration was determined in each group (Table 2). Although the dexamethasone group showed a slightly greater average HR slowing (3.7 bpm ± 3.8 vs 3.2 ± 3.9 in the control group), this difference was not statistically significant (p = 0.553, Fig 1). The maximal HR slowing was also not significant between groups (p = 0.988, Fig 1). After adjusting for baseline patient characteristics, the adjusted p-values were 0.570 and 0.907, respectively.”

4. I am unsure the sample size in this study is large enough to support or negate the original hypothesis, as the authors state in the Discussion.

Response: We thank the reviewer for the comment. The sample size for this study was determined based on logistical constraints, specifically the time period during which two medical students (VBK and MTA) were available to assist with data collection (May to September 2021). While the sample size was not chosen prospectively based on a formal power calculation for a specific primary outcome, we retrospectively computed the effect sizes that could be detected with adequate power (>80%) for the available sample size.

For binary outcomes, this sample size provides adequate power to detect differences as small as 30% (e.g., p1=35%, p2=65%; two-tailed alpha=0.05, group weights 2:1) using a chi-square test. For continuous outcomes, we computed that the sample size could detect effect sizes (Cohen’s d) as small as 0.6 with adequate power (>80%) using a two-sample t-test with the same alpha and group weights.

While the primary outcomes of interest were average HR change and maximal HR slowing, the retrospective calculation suggests that the study was not adequately powered to detect the smaller observed effect sizes (0.12 for HR change and 0.003 for max HR slowing). As such, the ability to detect significant differences for these specific outcomes was limited by the small observed effect sizes in our dataset. Additional detail about the sample size calculation were added under “Methods” (paragraph 3) and the limitations of our sample size were acknowledged under “Discussion” (paragraph 2).

Methods: Power and sample size calculation

“An a priori sample size calculation was not computed for this study. The sample size was determined based on a 3 month period (May to September 2021) when two medical students (VBK and MTA) were available to assist with data collection. After determining the available sample size, a retrospective analysis computed the effect sizes that could be detected with adequate power (>80%). It was determined that a sample size of 103 provided adequate power (>80%) to detect effect sizes as small as 30% for binary outcomes (e.g. p1=35%, p2=65%, two-tailed alpha=0.05, group weights 2:1 using the chi-square test). For continuous outcomes, this sample size provided adequate power (>80%) to detect effect sizes (Cohen’s d) as small as 0.6 between groups (two-sample t-test, two-tailed alpha=0.05, group weights 2:1).”

Discussion:

“Furthermore, the small sample size was a limitation of the study. The retrospective calculation suggested that this study was not adequately powered to detect the smaller observed effect sizes (0.12 for average HR change and 0.003 for max HR slowing). As such, the ability to detect significant differences for these specific outcomes was limited by the small observed effect sizes in our dataset.”

5. Please include the results of the secondary analyses (i.e. effect of peri-operative beta-blockers on bradycardia) in the Results section. How many participants experienced bradycardia after sugammedex? How was bradycardia defined?

Response: We thank the reviewer for this request. We evaluated the effect of perioperative beta-blockers (BB) on the incidence of bradycardia (defined as HR less than 60 bpm), given the potential mechanistic role of beta-blockers in modulating heart rate. We performed adjusted analyses to account for beta-blocker usage as well as other covariates listed in Table 1 (including race, age, BMI, ASA classification, hypertension, diabetes, coronary artery disease, malignancy, insulin use, and ACEI/ARBs).

The findings are as follows:

• Overall Cohort:

o Without perioperative beta-blockers: 38.3% (31/81) experienced bradycardia.

o With perioperative beta

---

## [Decision Letter · Decision Letter 1]

25 Feb 2025

PONE-D-24-42641R1Does Pre-emptive Dexamethasone Provide Prophylaxis Against Sugammadex-Induced Bradycardia? A Retrospective StudyPLOS ONE

Dear Dr. Jahr,

Thank you for submitting your manuscript to PLOS ONE. After careful consideration, we feel that it has merit but does not fully meet PLOS ONE’s publication criteria as it currently stands. Therefore, we invite you to submit a revised version of the manuscript that addresses the points raised during the review process.

We look forward to receiving your revised manuscript.

Kind regards,

Mohammed Misbah Ul Haq, Pharm-D

Academic Editor

PLOS ONE

Journal Requirements:

Reviewers' comments:

Reviewer's Responses to Questions

**Comments to the Author**

1. If the authors have adequately addressed your comments raised in a previous round of review and you feel that this manuscript is now acceptable for publication, you may indicate that here to bypass the “Comments to the Author” section, enter your conflict of interest statement in the “Confidential to Editor” section, and submit your "Accept" recommendation.

Reviewer #2: All comments have been addressed

2. Is the manuscript technically sound, and do the data support the conclusions?

Reviewer #2: Yes

3. Has the statistical analysis been performed appropriately and rigorously? 

Reviewer #2: Yes

4. Have the authors made all data underlying the findings in their manuscript fully available?

Reviewer #2: Yes

5. Is the manuscript presented in an intelligible fashion and written in standard English?

Reviewer #2: Yes

6. Review Comments to the Author

Reviewer #2: I am pleased to see the authors have addressed previous comments and appreciate the additional details that have been added to the manuscript. This paper will hopefully encourage further exploration into this interesting topic

7. PLOS authors have the option to publish the peer review history of their article (what does this mean? ). If published, this will include your full peer review and any attached files.

**Do you want your identity to be public for this peer review?** For information about this choice, including consent withdrawal, please see our Privacy Policy .

Reviewer #2: No

---

## [Author Response · Author response to Decision Letter 2]

17 Mar 2025

March 14, 2025

Mohammed Misbah Ul Haq, Pharm-D

RE: PLOS ONE Decision: Revision required [PONE-D-24-42641R1] - [EMID:e73e9f849082080b]

Dear Dr. Ul Haq:

Thank you for your acceptance of our manuscript listed above pending final revisions.

Based on our recent correspondence with Meghan Hom, we believe that we have addressed all the reviewers comments and completed the journal requirements. The authors reviewed the citations and none of them have been retracted, therefore not needing any replacements. Consequently, no changes were made to the list of references in our last submitted version, and we are resending that same version to you as our final.

Please confirm final acceptance of our manuscript.

Kindly reach out to me with any questions.

Sincerely,

Jonathan S. Jahr, MD, PhD, DABA, FASA

Professor Emeritus of Anesthesiology and Perioperative Medicine

Co-President, International Symposium on Blood Substitutes 2024 (isbs2024.org)

Chair of the Faculty, Emeritus (DGSOM Faculty Executive Committee)

David Geffen School of Medicine at UCLA

Ronald Reagan UCLA Medical Center

Jsjahr@mednet.ucla.edu; j.s.jahr@ucla.edu

Jonathan.jahr.2012@anderson.ucla.edu

Jonathan.jahr@VA.GOV

---

## [Decision Letter · Decision Letter 2]

8 Apr 2025

Does Pre-emptive Dexamethasone Provide Prophylaxis Against Sugammadex-Induced Bradycardia? A Retrospective Study

PONE-D-24-42641R2

Dear Dr. Jonathan S. Jahr,

We’re pleased to inform you that your manuscript has been judged scientifically suitable for publication and will be formally accepted for publication once it meets all outstanding technical requirements.

Kind regards,

Dr. Mohammed Misbah Ul Haq, Pharm-D

Academic Editor

PLOS ONE

Additional Editor Comments (optional):

Reviewers' comments:

Reviewer's Responses to Questions

**Comments to the Author**

1. If the authors have adequately addressed your comments raised in a previous round of review and you feel that this manuscript is now acceptable for publication, you may indicate that here to bypass the “Comments to the Author” section, enter your conflict of interest statement in the “Confidential to Editor” section, and submit your "Accept" recommendation.

Reviewer #2: All comments have been addressed

2. Is the manuscript technically sound, and do the data support the conclusions?

Reviewer #2: Yes

3. Has the statistical analysis been performed appropriately and rigorously? 

Reviewer #2: Yes

4. Have the authors made all data underlying the findings in their manuscript fully available?

Reviewer #2: Yes

5. Is the manuscript presented in an intelligible fashion and written in standard English?

Reviewer #2: Yes

6. Review Comments to the Author

Reviewer #2: I am pleased to see the authors have addressed previous comments and appreciate the additional details that have been added to the manuscript.

7. PLOS authors have the option to publish the peer review history of their article (what does this mean? ). If published, this will include your full peer review and any attached files.

**Do you want your identity to be public for this peer review?** For information about this choice, including consent withdrawal, please see our Privacy Policy .

Reviewer #2: No

---

## [Editor Report · Acceptance letter]

PONE-D-24-42641R2

PLOS ONE

Dear Dr. Jahr,

I'm pleased to inform you that your manuscript has been deemed suitable for publication in PLOS ONE. Congratulations! Your manuscript is now being handed over to our production team.

Kind regards,

on behalf of

Dr. Mohammed Misbah Ul Haq

Academic Editor

PLOS ONE